# A comparison of comorbidity measures for predicting mortality after elective hip and knee replacement: A cohort study of data from the National Joint Registry in England and Wales

**Chris M. Penfold** [1,2] *****, **Michael R. Whitehouse** [1,2], **Ashley W. Blom** [1,2], **Andrew Judge** [1,2], **J. Mark Wilkinson** [3,4], **Adrian Sayers** [1]

**1** Musculoskeletal Research Unit, Translational Health Sciences, Bristol Medical School, Southmead Hospital, Bristol, United Kingdom, **2** National Institute for Health Research Bristol Biomedical Research Centre, University Hospitals Bristol and Weston NHS Foundation Trust and University of Bristol, Bristol, United Kingdom, **3** Department of Oncology and Metabolism, University of Sheffield, Sorby Wing, Northern General Hospital, Sheffield, United Kingdom, **4** Centre for Integrated Research into Musculoskeletal Ageing, University of Sheffield, Sheffield, United Kingdom

***** chris.penfold@bristol.ac.uk

**Data Availability Statement:** Access to the data analysed in this study required permission from

## Abstract

### Background

The risk of mortality following elective total hip (THR) and knee replacements (KR) may be influenced by patients' pre-existing comorbidities. There are a variety of scores derived from individual comorbidities that can be used in an attempt to quantify this. The aims of this study were to a) identify which comorbidity score best predicts risk of mortality within 90 days or b) determine which comorbidity score best predicts risk of mortality at other relevant timepoints (30, 45, 120 and 365 days).

### Patients and methods

We linked data from the National Joint Registry (NJR) on primary elective hip and knee replacements performed between 2011–2015 with pre-existing conditions recorded in the Hospital Episodes Statistics. We derived comorbidity scores (Charlson Comorbidity Index—CCI, Elixhauser, Hospital Frailty Risk Score—HFRS). We used binary logistic regression models of all-cause mortality within 90-days and within 30, 45, 120 and 365-days of the primary operation using, adjusted for age and gender. We compared the performance of these models in predicting all-cause mortality using the area under the Receiver-operator characteristics curve (AUROC) and the Index of Prediction Accuracy (IPA).

### Results

We included 276,594 elective primary THRs and 338,287 elective primary KRs for any indication. Mortality within 90-days was 0.34% (N = 939) after THR and 0.26% (N = 865) after

the National Joint Registry for England, Wales and Northern Ireland Research Sub-committee. http://www.njrcentre.org.uk/njrcentre/Research/Researchrequests/tabid/305/Default.aspx contains information on research data access request to the National Joint Registry.

**Funding:** CP, AB, MRW and AJ acknowledge support by the NIHR Biomedical Research Centre at University Hospitals Bristol NHS Foundation Trust and the University of Bristol (https://www.bristolbrc.nihr.ac.uk/). The views expressed in this publication are those of the authors and not necessarily those of the NHS, the National Institute for Health Research or the Department of Health and Social Care, the National Joint Registry Steering Committee or Healthcare Quality Improvement Partnership, who do not vouch for how the information is presented. AS was supported by a MRC fellowship MR/L01226X/1. AS was supported by a contract grant from the National Joint Registry for England, Wales, Northern Ireland and the Isle of Man (https://www.njrcentre.org.uk/njrcentre/default.aspx). The funders had no role in study design, data collection and analysis, decision to publish, or preparation of the manuscript.

**Competing interests:** I have read the journal's policy and the authors of this manuscript have the following competing interests: MW (Stryker, Heraeus, DePuy), AB (Stryker) and JMW (Amgen) have received research and other financial support from companies or suppliers outside the submitted work. AJ declares advisory board positions with receipt of fees (Anthera Pharmaceuticals, INC.) and paid consultancy work (Freshfields Bruckhaus Deringer) for companies outside the submitted work. MRW (Hip International) and JMW (Bone and Joint Research, Journal of Orthopaedic Research) declare journal editorial positions. JMW is a board member for the British Orthopaedic Research Society. All other authors declare no competing interests. This does not alter our adherence to PLOS ONE policies on sharing data and materials.

**Abbreviations:** ASA grade, American Society of Anaesthesiologists classification; AUROC, Area Under the Receiver-Operator Characteristics curve; CCI, Charlson Comorbidity Index; CPRD, Clinical Practice Research Database; HES, Hospital Episodes Statistics; HFRS, Hospital Frailty Risk Score; ICD-10, International Classification of Diseases (10th Edition); KR, Knee Replacement; NJR, National Joint Registry for England, Wales, Northern Ireland, the Isle of Man and the States of Guernsey; ROC curve, Receiver-Operator Characteristics curve; SHMI, Summary Hospital-

KR. The AUROC for the CCI and Elixhauser scores in models of mortality ranged from 0.78–0.81 after THR and KR, which slightly outperformed models with ASA grade (AUROC = 0.77–0.78). HFRS performed similarly to ASA grade (AUROC = 0.76–0.78). The inclusion of comorbidities prior to the primary operation offers no improvement beyond models with comorbidities at the time of the primary. The discriminative ability of all prediction models was best for mortality within 30 days and worst for mortality within 365 days.

## Conclusions

Comorbidity scores add little improvement beyond simpler models with age, gender and ASA grade for predicting mortality within one year after elective hip or knee replacement. The additional patient-specific information required to construct comorbidity scores must be balanced against their prediction gain when considering their utility.

## Background

Elective knee (KR) and total hip (THR) replacement are amongst the most commonly performed elective operations. They are also highly successful procedures with typical 10-year revision rates of <5% [1]. Mortality after primary hip and knee replacement is rare and has decreased in recent years [2, 3]. The National Joint Registry for England, Wales, Northern Ireland, the Isle of Man and the States of Guernsey (NJR) routinely monitors mortality outcomes at surgeon and unit level. This process includes case-mix adjustment for age, gender, indication for surgery and American Society of Anaesthesiologists physical status (ASA grade) which records the preoperative health of surgical patients.

The presence of comorbidities (pre-existing health conditions that coexist with an index disease) is associated with worse health outcomes and more complex clinical management [4]. Comorbidity has been found to be a predictor of perioperative and in-hospital mortality [5], and a risk factor for 90-day mortality after joint replacement [6]. The use of comorbidity score in place of ASA grade may improve prediction of mortality risk, but collection of comorbidities is much more complex and laborious than ASA grade.

Many summary indices of comorbidities based on diagnoses have been derived, however the main focus within replacement surgery has been on the Charlson Comorbidity (CCI) and Elixhauser indices [7]. The Elixhauser index includes 30 conditions and is a composite measurement to assess the impact of comorbidity on surgical procedures [8] and the CCI includes 17 conditions [9]. The Elixhauser index predicted inpatient mortality after orthopaedic surgery better than the CCI [5]. However, comorbidity does not predict long-term mortality [10]. Recent developments in comorbidity scores include the Hospital Frailty Risk Score (HFRS) [11], designed to screen for frailty and identify a group of patients who are at greater risk of adverse outcomes. This was found to predict adverse events after THRs and KRs, but its performance was not compared against other comorbidity indices [12].

The aims of this study are:

1. To determine which comorbidity score best predicts risk of mortality within 90 days of elective primary hip and knee replacement

2. To determine which comorbidity score best predicts risk of mortality within other landmark postoperative timepoints (30, 45, 120, 365 days) after elective primary hip and knee replacement

level Mortality Indicator; THR, Total Hip Replacement.

## Methods

### Data source

The National Joint Registry (NJR) was established in 2003 [1]. The NJR includes nearly 2.5 million primary THRs and KRs in patients aged >18 years performed in public and private hospitals in England, Wales, Northern Ireland, the Isle of Man and the States of Guernsey [13]. Data are collected at the time of surgery on prosthesis and operative information, patient information, and surgical and unit information. We linked these records to Hospital Episodes Statistics (HES)–Admitted Patient Care data, established in 1989 [14], for all available episodes up to and including the primary joint replacement operation. For people who had contralateral primary operations we linked separate HES records for each primary operation. Date of death was linked at the person-level using civil registration mortality records.

### Ethics approval and consent to participate

Patient consent was obtained for data collection by the NJR. According to the specifications of the NHS Health Research Authority, separate informed consent and ethical approval were not required for the present study.

### Study sample

We included patients who received a primary elective THR or KR for any indication between January 1st 2011 and 31st December 2015. Patients were followed up until 31st December 2016. We only included primary operations that could be linked to HES records. This excluded privately funded operations since these episodes are not recorded in HES and hence comorbidity indices could not be derived at the time of the primary operation. This also excluded operations performed in Wales and Scotland, since HES data collection only covers operations performed in England. We excluded primary operations performed in Northern Ireland, the Isle of Man and Guernsey, since data collection in these regions only commenced in 2013, 2015, and 2020, respectively. We also excluded people who had not given consent for recording of personal details for research purposes and primary operations performed for trauma (see Figs 1 & 2).

### Potential predictors

The patient's age at the time of surgery (in years, natural spline with knot points at 50 and 75 years) and gender (categorical) were included as potential predictors in all models and comprised our base model. Our reference model contained ASA grade (categorical predictor: 'I', 'II', 'III', 'IV & V'), which is routinely recorded in the NJR and has the advantage of not requiring linkage to other datasets. We used pre-existing conditions recorded in HES using ICD-10 codes to derive the following comorbidity scores:

- CCI with two weightings:

  ○ Original weightings by Charlson et al. [9]: categorised '0', '1', '2', '3+'

  ○ Revised weightings developed by Dr Foster Intelligence and used by the Health and Social Care Information Centre as part of the Summary Hospital-level Mortality Indicator (SHMI) [15]: continuous variable

- Elixhauser comorbidity index [8]: continuous variable

- Hospital Frailty Risk Score (HFRS) [11]: continuous variable

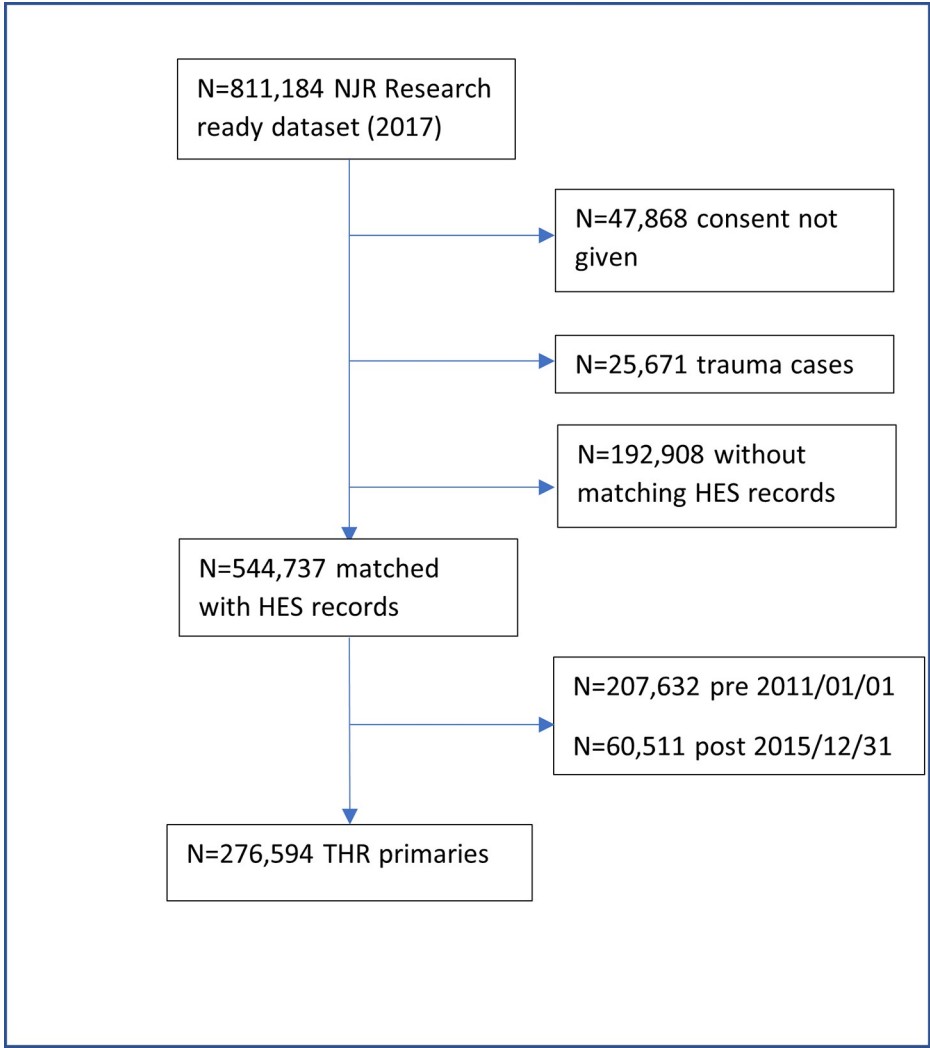

**Fig 1. Flowchart of eligible primary THRs.**

We derived the comorbidity scores using pre-existing conditions recorded over the following timeframes:

- At the time of the primary operation

- Any episodes in the 1 year up to the primary

- Any episodes in the 2 years up to the primary

- Any episodes in the 5 years up to the primary

- All available episodes up to the primary

## Outcomes

**All-cause mortality.** Our primary outcome was mortality from all causes within 90 days of the primary operation. Secondary outcomes were all-cause mortality within 30, 45, 120 and 365 days of the primary operation.

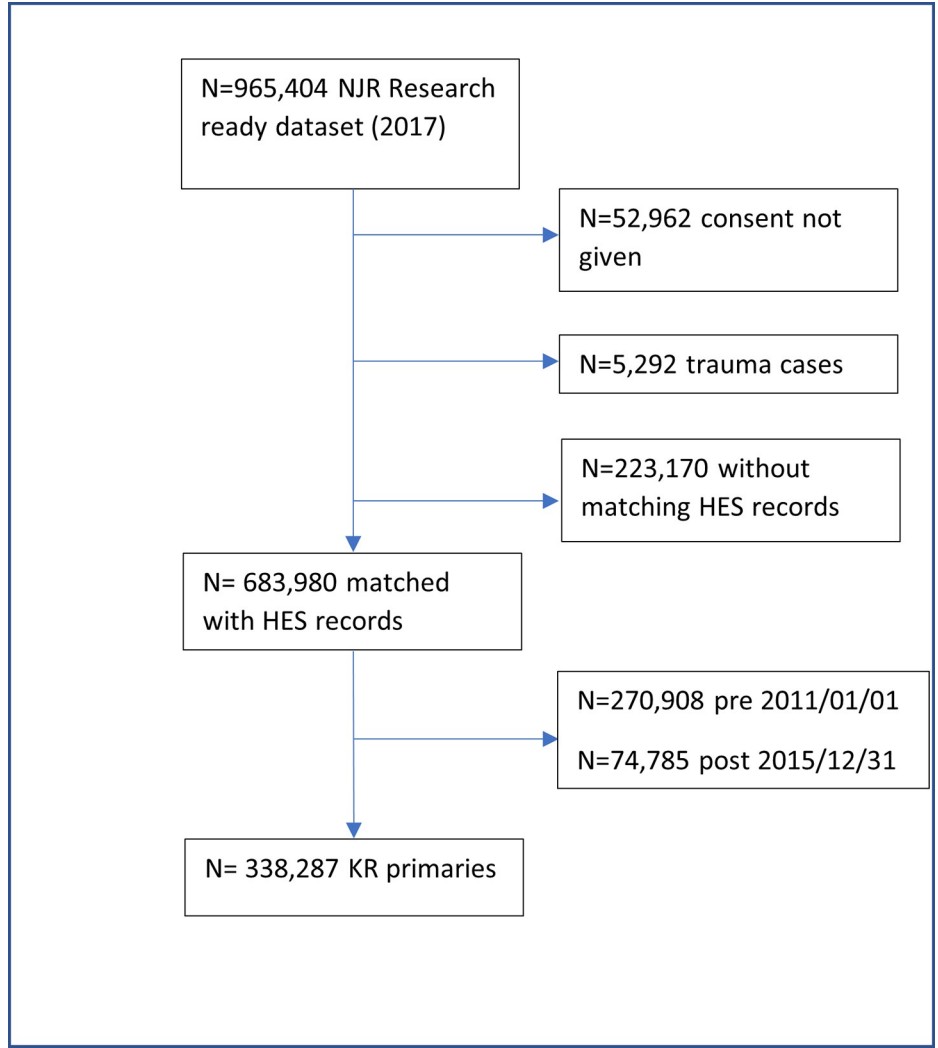

**Fig 2. Flowchart of eligible primary KRs.**

## Statistical analysis

We analysed mortality outcomes for primary hip and knee replacements separately. We described the comorbidity of people undergoing elective THR and KR operations. We used predicted probabilities from logistic regression models to identify the best comorbidity predictors of mortality, and the optimal timeframe over which to define the comorbidity scores. We constructed the following regression models:

- Base model: Age and gender

- Model 1 (reference): Base + ASA

- Model 2: Base + CCI (original)

- Model 3: Base + CCI (SHMI)

- Model 4: Base + Elixhauser

- Model 5: Base + HFRS

We compared models using area under the receiver operating characteristic curve (AUROC), a measure of how well the model discriminates between those who experience the outcome and those who don't (values 0 to 1, 0.5 = no discrimination, higher value = better classifier), and the Index of Prediction Accuracy (IPA) [16], calculated from the null model and model Brier scores to combine discrimination and calibration in a single value (values -1 to 1, 1 is a perfect model, <0 is a harmful model). We performed internal validation using 5-fold cross-validation, and reported the overall results and results of our primary analyses for each fold.

**Sensitivity analyses.**   Some patients received a second primary THR or KR on the opposite joint (contralateral primary) within our follow-up timeframe. These patients will contribute twice to our analyses. We therefore excluded the earliest performed primary and repeated our main analyses.

Data were processed in Stata v15 (StataCorp) and all analyses were performed using R version 4.0 [17] and the 'tidymodels' packages [18]. Confidence intervals (95% CI) were derived using the exact method to evaluate the uncertainty of AUC developed by DeLong [19] and implemented using the algorithm proposed by Sun and Xu [20] in the 'pROC' package [21].

## Results

Our study sample was 276,594 primary THRs and 338,287 primary KRs which met the inclusion criteria (Figs 1 and 2). The proportion of people who died within 90 days was 0.34% (N = 954) after THR (Table 1) and 0.26% (N = 870) after KR (Table 2). Secondary mortality timepoints after THR were: 30 days 0.17% (n = 465), 45 days 0.21% (n = 592), 120 days 0.43% (n = 1,187), 365 days 1.20% (n = 3,314); and after KR were: 30 days 0.14% (n = 470), 45 days 0.17% (n = 578), 120 days 0.31% (n = 1,064), 365 days 0.91% (n = 3,085).

In patients who died within 90 days of their primary operation the five most prevalent comorbidities from the CCI were very similar for people who had a THR or KR: COPD (THR 20%, KR 22%), diabetes without complications (THR 17%, KR 22%), renal disease (THR 16%, KR 17%), acute myocardial infarction (THR and KR 10%) and congestive heart failure (THR 11%, KR 9.7%) (Tables 1 and 2). In the same patients, the most prevalent comorbidities from the Elixhauser index were very similar: uncomplicated hypertension (THR 52%, KR 61%), arrhythmia (THR 25%, KR 24%), chronic pulmonary disease (THR 20%, KR 22%), diabetes without complications (THR 17%, KR 21%) and renal failure (THR 16%, KR 17%). There was a marked difference in the prevalence of metastatic cancer between people who died within 90 days of their THR and KR: 12% and 0.8% respectively. This likely reflects the prophylactic replacement of the hip in patients with metastasis in the proximal femur to prevent a femoral fracture. Metastases in the distal femur, which may require a prophylactic knee replacement, occur much less frequently.

A comparison of comorbidity scores derived from varying lead-up times with those derived from all available episodes (S1–S4 Figs) highlights differences in the capture of high comorbidity scores. The majority of patients had CCI score 0 and the median comorbidity score for all measures at all time points, apart from HFRS derived using 5-year lead-up and all episodes, was 0 (S1 Table). Increasing the timeframe for deriving comorbidity scores decreased the proportion of patients with CCI = 0 and increased the comorbidity scores of the upper quartile a modest amount and the maximum comorbidity scores considerably.

## Comparison of models

**1. Comorbidity indices using comorbidities at time of primary.**   The AUROC indicate that, using comorbidities recorded at the time of the primary operation, the CCI (original and

**Table 1. The characteristics of people having a primary THR, including by 90-day mortality.**

| Characteristic | Alive at 90 days | Died by 90 days | Total |
|---|---|---|---|
| | N = 275,640[1] | N = 954[1] | N = 276,594[1] |
| **Patient age at surgery** | 70 (62, 76) | 78 (71, 83) | 70 (62, 76) |
| **Gender** | | | |
| Male | 109,303 (40%) | 501 (53%) | 109,804 (40%) |
| Female | 166,337 (60%) | 453 (47%) | 166,790 (60%) |
| **ASA Grade** | | | |
| I | 36,089 (13%) | 30 (3.1%) | 36,119 (13%) |
| II | 193,934 (70%) | 424 (44%) | 194,358 (70%) |
| III | 44,379 (16%) | 442 (46%) | 44,821 (16%) |
| IV +V | 1,238 (0.4%) | 58 (6.1%) | 1,296 (0.5%) |
| **Charlson comorbidities** | | | |
| Acute Myocardial Infarction | 7,202 (2.6%) | 95 (10.0%) | 7,297 (2.6%) |
| Congestive heart failure | 3,143 (1.1%) | 108 (11%) | 3,251 (1.2%) |
| Peripheral Vascular disease | 2,931 (1.1%) | 38 (4.0%) | 2,969 (1.1%) |
| Cerebrovascular disease | 1,730 (0.6%) | 39 (4.1%) | 1,769 (0.6%) |
| Dementia | 1,225 (0.4%) | 23 (2.4%) | 1,248 (0.5%) |
| Chronic Obstructive Pulmonary disease | 37,264 (14%) | 190 (20%) | 37,454 (14%) |
| Rheumatoid Disease | 10,225 (3.7%) | 46 (4.8%) | 10,271 (3.7%) |
| Peptic Ulcer | 360 (0.1%) | 8 (0.8%) | 368 (0.1%) |
| Mild liver disease | 1,148 (0.4%) | 12 (1.3%) | 1,160 (0.4%) |
| Diabetes | 25,860 (9.4%) | 159 (17%) | 26,019 (9.4%) |
| Diabetes + Complications | 673 (0.2%) | 10 (1.0%) | 683 (0.2%) |
| Hemiplegia or Paraplegia | 442 (0.2%) | 10 (1.0%) | 452 (0.2%) |
| Renal disease | 12,574 (4.6%) | 157 (16%) | 12,731 (4.6%) |
| Cancer | 2,951 (1.1%) | 114 (12%) | 3,065 (1.1%) |
| Moderate/Severe liver disease | 90 (<0.1%) | 5 (0.5%) | 95 (<0.1%) |
| Metastatic Cancer | 671 (0.2%) | 113 (12%) | 784 (0.3%) |
| AIDS | 0 (0%) | 0 (0%) | 0 (0%) |
| **Elixhauser comorbidities** | | | |
| Congestive Heart Failure | 3,143 (1.1%) | 108 (11%) | 3,251 (1.2%) |
| Cardiac Arrhythmias | 21,695 (7.9%) | 241 (25%) | 21,936 (7.9%) |
| Valvular Disease | 5,739 (2.1%) | 62 (6.5%) | 5,801 (2.1%) |
| Pulmonary Circulation Disorders | 719 (0.3%) | 32 (3.4%) | 751 (0.3%) |
| Peripheral Vascular Disorders | 2,931 (1.1%) | 38 (4.0%) | 2,969 (1.1%) |
| Hypertension, Uncomplicated | 124,027 (45%) | 496 (52%) | 124,523 (45%) |
| Paralysis | 442 (0.2%) | 10 (1.0%) | 452 (0.2%) |
| Other Neurological Disorders | 4,311 (1.6%) | 33 (3.5%) | 4,344 (1.6%) |
| Chronic Pulmonary Disease | 37,264 (14%) | 190 (20%) | 37,454 (14%) |
| Diabetes, Uncomplicated | 25,840 (9.4%) | 159 (17%) | 25,999 (9.4%) |
| Diabetes, Complicated | 686 (0.2%) | 10 (1.0%) | 696 (0.3%) |
| Hypothyroidism | 18,948 (6.9%) | 54 (5.7%) | 19,002 (6.9%) |
| Renal Failure | 12,567 (4.6%) | 156 (16%) | 12,723 (4.6%) |
| Liver Disease | 1,172 (0.4%) | 20 (2.1%) | 1,192 (0.4%) |
| Peptic Ulcer Disease Excluding Bleeding | 322 (0.1%) | 3 (0.3%) | 325 (0.1%) |
| AIDS/HIV | 0 (0%) | 0 (0%) | 0 (0%) |
| Lymphoma | 459 (0.2%) | 7 (0.7%) | 466 (0.2%) |
| Metastatic Cancer | 671 (0.2%) | 113 (12%) | 784 (0.3%) |

*(Continued)*

**Table 1.** (Continued)

| Characteristic | Alive at 90 days | Died by 90 days | Total |
|---|---|---|---|
| | N = 275,640[1] | N = 954[1] | N = 276,594[1] |
| Solid Tumor Without Metastasis | 2,095 (0.8%) | 102 (11%) | 2,197 (0.8%) |
| Rheumatoid Arthritis/Collagen Vascular | 12,020 (4.4%) | 51 (5.3%) | 12,071 (4.4%) |
| Coagulopathy | 1,052 (0.4%) | 11 (1.2%) | 1,063 (0.4%) |
| Obesity | 27,689 (10%) | 68 (7.1%) | 27,757 (10%) |
| Weight Loss | 103 (<0.1%) | 1 (0.1%) | 104 (<0.1%) |
| Fluid and Electrolyte Disorders | 3,724 (1.4%) | 110 (12%) | 3,834 (1.4%) |
| Blood Loss Anemia | 104 (<0.1%) | 1 (0.1%) | 105 (<0.1%) |
| Deficiency Anemia | 2,127 (0.8%) | 15 (1.6%) | 2,142 (0.8%) |
| Alcohol Abuse | 5,065 (1.8%) | 24 (2.5%) | 5,089 (1.8%) |
| Drug Abuse | 314 (0.1%) | 2 (0.2%) | 316 (0.1%) |
| Psychoses | 382 (0.1%) | 7 (0.7%) | 389 (0.1%) |
| Depression | 10,862 (3.9%) | 36 (3.8%) | 10,898 (3.9%) |
| Hypertension, Complicated | 1,228 (0.4%) | 17 (1.8%) | 1,245 (0.5%) |

[1]Median (IQR); n (%).

SHMI, $AUROC_{THR}$ = 0.80 and $AUROC_{KR}$ = 0.78) and Elixhauser scores ($AUROC_{THR}$ = 0.81 and $AUROC_{KR}$ = 0.78) slightly outperformed ASA grade ($AUROC_{THR}$ = 0.78 and $AUROC_{KR}$ = 0.77) in predicting 90-day mortality after THR and KR (Table 3 and Figs 3 and 4). HFRS performed similarly to ASA grade in predicting 90-day mortality after THR and KR ($AUROC_{THR}$ = 0.77, $AUROC_{KR}$ = 0.78). All models performed better than the base model (age and gender only, $AUROC_{THR}$ = 0.72, $AUROC_{KR}$ = 0.74). IPA scores for all models with comorbidity predictors recorded at the time of the primary were comparable or higher than models with ASA grade for THRs (IPA = 0.66% to 2.1% versus IPA = 0. 67%) and KRs (IPA = 0.51% to 1.0% versus IPA = 0.56%) and higher than those for the base models ($IPA_{THR}$ = 0.36%, $IPA_{KR}$ = 0.38%). ROC curves using comorbidity scores derived from conditions recorded at the time of the primary are shown in Figs 5 and 6.

**2. Comorbidity indices using history of comorbidities.** There was little difference between the discriminative abilities of comorbidity scores derived over different timeframes. The AUROC varied by a maximum of $1/10^{th}$ of a percentage point (Table 3). ROC curves for all timeframes are shown in S5–S12 Figs. IPA scores for all models with comorbidity predictors were highest when derived using comorbidities recorded at the time of the primary compared with those which were derived longer timeframes (Table 3). IPA scores for the CCI (original) and Elixhauser index were lowest when all available episodes were included, whereas IPA scores for CCI (SHMI) and HFRS were lowest when two to five years of preceding episodes were used to derive the scores.

**3. Landmarks.** For all comorbidity scores the performance of the prediction models after THR and KR was best for the shortest timeframe (30 days) and their performance worsened with increasing time (Table 4). For THAs, CCI (original and SHMI) and Elixhauser had marginally better discriminative ability than ASA. HFRS had better discriminative ability than ASA for mortality by 30 and 45 days, but was slightly worse for mortality by 120 and 365 days. For KRs, there was almost no difference in the discriminative ability of models with ASA grade compared with any comorbidity score, irrespective of the mortality timeframe. IPA scores increased with increasing time for all potential predictors, indicating improved accuracy for mortality predictions at one year compared with 30 days.

**Table 2. The characteristics of people having a primary KR, including by 90-day mortality.**

| Characteristic | Alive at 90 days | Died by 90 days | Total |
|---|---|---|---|
| | N = 337,417[1] | N = 870[1] | N = 338,287[1] |
| Patient age at surgery | 69 (63, 76) | 78 (72, 83) | 69 (63, 76) |
| **Gender** | | | |
| Male | 144,609 (43%) | 485 (56%) | 145,094 (43%) |
| Female | 192,808 (57%) | 385 (44%) | 193,193 (57%) |
| **ASA Grade** | | | |
| I | 31,924 (9.5%) | 23 (2.6%) | 31,947 (9.4%) |
| II | 249,274 (74%) | 485 (56%) | 249,759 (74%) |
| III | 55,227 (16%) | 335 (39%) | 55,562 (16%) |
| IV +V | 992 (0.3%) | 27 (3.1%) | 1,019 (0.3%) |
| **Charlson comorbidities** | | | |
| Acute Myocardial Infarction | 8,494 (2.5%) | 91 (10%) | 8,585 (2.5%) |
| Congestive heart failure | 3,099 (0.9%) | 84 (9.7%) | 3,183 (0.9%) |
| Peripheral Vascular disease | 3,199 (0.9%) | 33 (3.8%) | 3,232 (1.0%) |
| Cerebrovascular disease | 2,171 (0.6%) | 38 (4.4%) | 2,209 (0.7%) |
| Dementia | 1,123 (0.3%) | 14 (1.6%) | 1,137 (0.3%) |
| Chronic Obstructive Pulmonary disease | 49,782 (15%) | 189 (22%) | 49,971 (15%) |
| Rheumatoid Disease | 14,822 (4.4%) | 62 (7.1%) | 14,884 (4.4%) |
| Peptic Ulcer | 539 (0.2%) | 7 (0.8%) | 546 (0.2%) |
| Mild liver disease | 1,269 (0.4%) | 14 (1.6%) | 1,283 (0.4%) |
| Diabetes | 43,851 (13%) | 184 (21%) | 44,035 (13%) |
| Diabetes + Complications | 1,060 (0.3%) | 8 (0.9%) | 1,068 (0.3%) |
| Hemiplegia or Paraplegia | 536 (0.2%) | 7 (0.8%) | 543 (0.2%) |
| Renal disease | 15,030 (4.5%) | 149 (17%) | 15,179 (4.5%) |
| Cancer | 2,877 (0.9%) | 18 (2.1%) | 2,895 (0.9%) |
| Moderate/Severe liver disease | 67 (<0.1%) | 9 (1.0%) | 76 (<0.1%) |
| Metastatic Cancer | 224 (<0.1%) | 7 (0.8%) | 231 (<0.1%) |
| AIDS | 0 (0%) | 0 (0%) | 0 (0%) |
| **Elixhauser comorbidities** | | | |
| Congestive Heart Failure | 3,099 (0.9%) | 84 (9.7%) | 3,183 (0.9%) |
| Cardiac Arrhythmias | 25,624 (7.6%) | 205 (24%) | 25,829 (7.6%) |
| Valvular Disease | 6,108 (1.8%) | 57 (6.6%) | 6,165 (1.8%) |
| Pulmonary Circulation Disorders | 1,318 (0.4%) | 27 (3.1%) | 1,345 (0.4%) |
| Peripheral Vascular Disorders | 3,199 (0.9%) | 33 (3.8%) | 3,232 (1.0%) |
| Hypertension, Uncomplicated | 172,530 (51%) | 534 (61%) | 173,064 (51%) |
| Paralysis | 536 (0.2%) | 7 (0.8%) | 543 (0.2%) |
| Other Neurological Disorders | 6,277 (1.9%) | 28 (3.2%) | 6,305 (1.9%) |
| Chronic Pulmonary Disease | 49,782 (15%) | 189 (22%) | 49,971 (15%) |
| Diabetes, Uncomplicated | 43,806 (13%) | 184 (21%) | 43,990 (13%) |
| Diabetes, Complicated | 1,098 (0.3%) | 8 (0.9%) | 1,106 (0.3%) |
| Hypothyroidism | 25,139 (7.5%) | 59 (6.8%) | 25,198 (7.4%) |
| Renal Failure | 15,023 (4.5%) | 149 (17%) | 15,172 (4.5%) |
| Liver Disease | 1,301 (0.4%) | 20 (2.3%) | 1,321 (0.4%) |
| Peptic Ulcer Disease Excluding Bleeding | 494 (0.1%) | 3 (0.3%) | 497 (0.1%) |
| AIDS/HIV | 0 (0%) | 0 (0%) | 0 (0%) |
| Lymphoma | 374 (0.1%) | 3 (0.3%) | 377 (0.1%) |
| Metastatic Cancer | 224 (<0.1%) | 7 (0.8%) | 231 (<0.1%) |
| Solid Tumor Without Metastasis | 2,071 (0.6%) | 14 (1.6%) | 2,085 (0.6%) |
| Rheumatoid Arthritis/Collagen Vascular | 16,361 (4.8%) | 66 (7.6%) | 16,427 (4.9%) |
| Coagulopathy | 1,254 (0.4%) | 14 (1.6%) | 1,268 (0.4%) |

*(Continued)*

**Table 2.** (Continued)

| Characteristic | Alive at 90 days | Died by 90 days | Total |
|---|---|---|---|
| | N = 337,417[1] | N = 870[1] | N = 338,287[1] |
| Obesity | 48,727 (14%) | 80 (9.2%) | 48,807 (14%) |
| Weight Loss | 40 (<0.1%) | 2 (0.2%) | 42 (<0.1%) |
| Fluid and Electrolyte Disorders | 4,449 (1.3%) | 87 (10%) | 4,536 (1.3%) |
| Blood Loss Anemia | 73 (<0.1%) | 2 (0.2%) | 75 (<0.1%) |
| Deficiency Anemia | 2,723 (0.8%) | 18 (2.1%) | 2,741 (0.8%) |
| Alcohol Abuse | 5,606 (1.7%) | 18 (2.1%) | 5,624 (1.7%) |
| Drug Abuse | 177 (<0.1%) | 1 (0.1%) | 178 (<0.1%) |
| Psychoses | 362 (0.1%) | 2 (0.2%) | 364 (0.1%) |
| Depression | 13,918 (4.1%) | 33 (3.8%) | 13,951 (4.1%) |
| Hypertension, Complicated | 1,571 (0.5%) | 22 (2.5%) | 1,593 (0.5%) |

[1]Median (IQR); n (%).

**Table 3.** The area under the ROC curve and IPA for ASA grade and all comorbidity scores for models of 90-day mortality after THR and KR, adjusted for age and gender.

| Model | THRs | | | KRs | | |
|---|---|---|---|---|---|---|
| | AUROC[1] | 95% CI[2] | IPA[3] | AUROC[1] | 95% CI[2] | IPA[3] |
| **Base** | 0.720 | 0.703–0.737 | 0.0036 | 0.744 | 0.727–0.761 | 0.0038 |
| **ASA** | 0.775 | 0.760–0.791 | 0.0067 | 0.766 | 0.750–0.782 | 0.0056 |
| **CCI (original)** | | | | | | |
| Primary episode | 0.802 | 0.788–0.816 | 0.0087 | 0.775 | 0.759–0.792 | 0.0069 |
| 1-year lead-up | 0.801 | 0.787–0.815 | 0.0082 | 0.775 | 0.759–0.792 | 0.0065 |
| 2-year lead-up | 0.803 | 0.789–0.817 | 0.0082 | 0.776 | 0.760–0.792 | 0.0062 |
| 5-year lead-up | 0.799 | 0.785–0.812 | 0.0074 | 0.774 | 0.758–0.791 | 0.0062 |
| All episodes | 0.792 | 0.778–0.806 | 0.0070 | 0.774 | 0.757–0.790 | 0.0057 |
| **CCI (SHMI)** | | | | | | |
| Primary episode | 0.802 | 0.788–0.817 | 0.0088 | 0.781 | 0.765–0.797 | 0.0098 |
| 1-year lead-up | 0.799 | 0.785–0.814 | 0.0060 | 0.780 | 0.764–0.797 | 0.0088 |
| 2-year lead-up | 0.799 | 0.784–0.813 | 0.0053 | 0.779 | 0.763–0.795 | 0.0070 |
| 5-year lead-up | 0.792 | 0.777–0.806 | 0.0059 | 0.780 | 0.764–0.796 | 0.0074 |
| All episodes | 0.786 | 0.771–0.800 | 0.0060 | 0.778 | 0.762–0.794 | 0.0071 |
| **Elixhauser** | | | | | | |
| Primary episode | 0.808 | 0.794–0.823 | 0.0214 | 0.779 | 0.762–0.795 | 0.0103 |
| 1-year lead-up | 0.810 | 0.796–0.825 | 0.0141 | 0.779 | 0.762–0.795 | 0.0085 |
| 2-year lead-up | 0.815 | 0.802–0.829 | 0.0099 | 0.780 | 0.764–0.796 | 0.0074 |
| 5-year lead-up | 0.814 | 0.801–0.828 | 0.0080 | 0.778 | 0.761–0.794 | 0.0068 |
| All episodes | 0.810 | 0.796–0.824 | 0.0078 | 0.774 | 0.758–0.791 | 0.0062 |
| **Frailty** | | | | | | |
| Primary episode | 0.765 | 0.748–0.782 | 0.0066 | 0.777 | 0.760–0.793 | 0.0051 |
| 1-year lead-up | 0.769 | 0.753–0.786 | 0.0034 | 0.775 | 0.759–0.792 | 0.0024 |
| 2-year lead-up | 0.769 | 0.753–0.785 | 0.0027 | 0.775 | 0.758–0.791 | 0.0018 |
| 5-year lead-up | 0.766 | 0.750–0.783 | 0.0026 | 0.773 | 0.756–0.790 | 0.0028 |
| All episodes | 0.763 | 0.747–0.780 | 0.0036 | 0.771 | 0.755–0.788 | 0.0033 |

1 –AUROC–area under the ROC curve.

2–95% CI– 95% confidence intervals.

3 –IPA–Index of Prediction Accuracy.

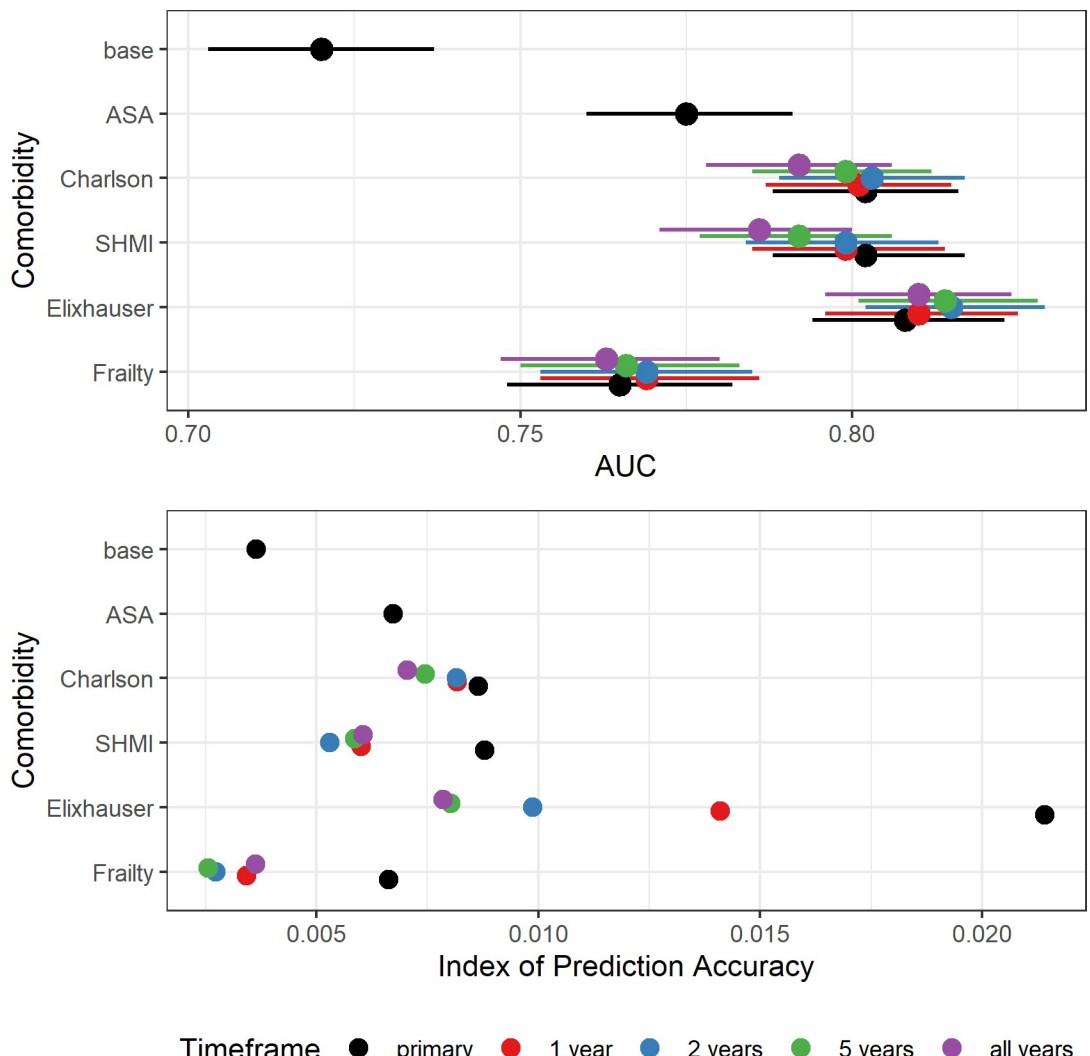

**Fig 3. Area under the ROC curves (higher better, lines represent 95% CIs) and IPA scores (higher better) for models of 90-day mortality after THR with ASA grade and the 4 comorbidity indices for all time frames, including age and gender.**

## Sensitivity analyses

The results from the five-fold cross-validation show variability of approximately five to seven percentage points in the $AUC_{THR}$ and approximately two and four percentage points in the $AUC_{KR}$ between the best and worst performing folds (S4 and S5 Tables). IPA scores varied considerably, including with two of the five folds indicating harmful models (negative IPA scores).

We excluded 12,723 contralateral THRs and 20,703 contralateral KRs performed within one year of the corresponding first primary operation. Results of our primary analyses changed by only 0.3 percentage points (results not reported).

## Discussion

We compared the performance of four comorbidity scores (CCI with original and SHMI weights, Elixhauser Index and HFRS) in predicting the risk of all-cause mortality within 30,

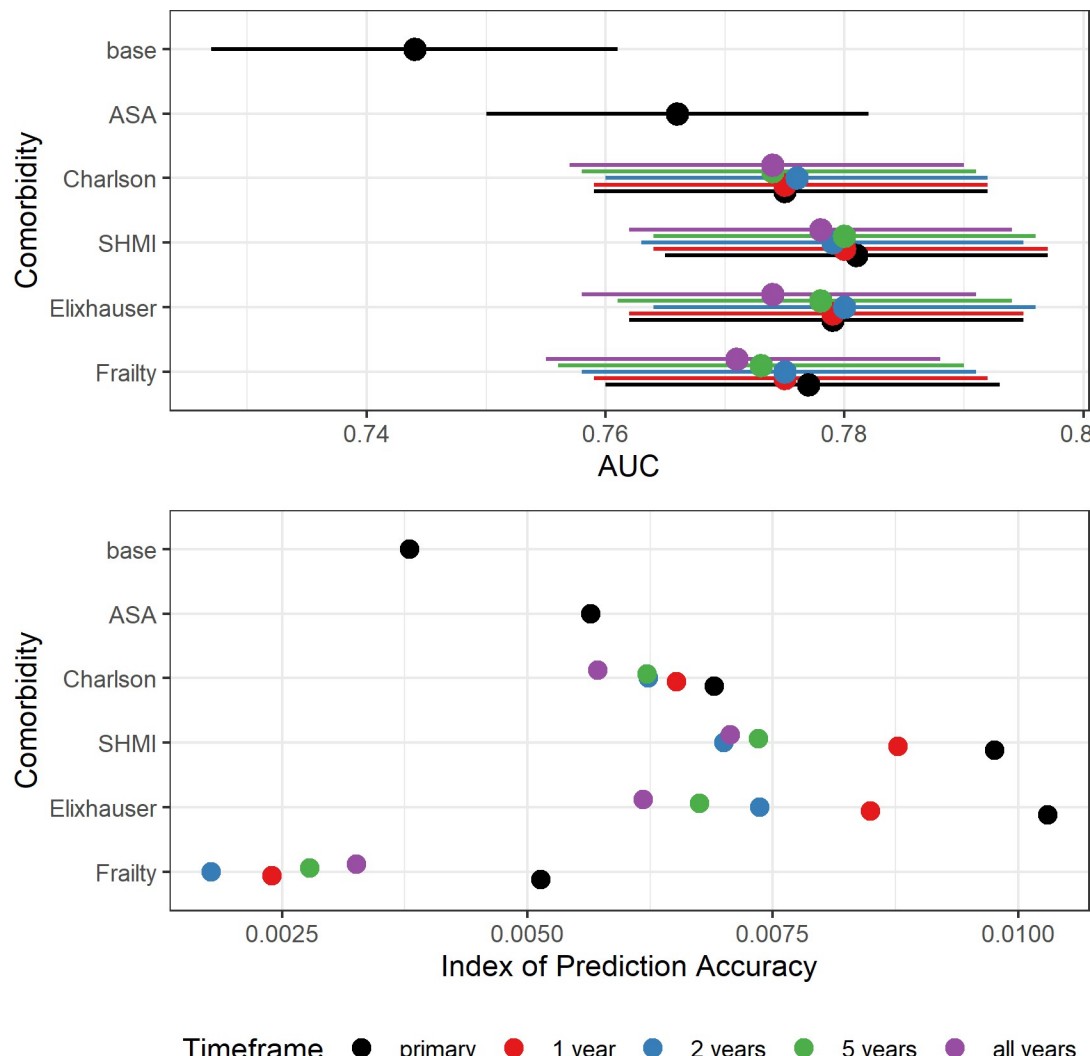

**Fig 4. Area under the ROC curves (higher better, lines represent 95% CIs) and IPA scores (higher better) for models of 90-day mortality after KR with ASA grade and the 4 comorbidity indices for all time frames, including age and gender.**

45, 90, 120 and 365 days of primary elective THRs and KRs. We found that mortality predictions from models with comorbidity scores add only a modest improvement compared with those from models with ASA grade. The CCI (original and SHMI) and Elixhauser scores all performed slightly better than ASA grade in predicting mortality after THR. The inclusion of comorbidities either at the time of or prior to the primary operation offers little improvement beyond models with ASA grade in the prediction of the risk of dying up to one year.

The main strengths of this study relate to the size and completeness of the NJR dataset, and the HES linkage. Mortality within 90 days of elective hip or knee replacement is a rare event and remains so up to one year after the primary operation. The size of the NJR meant that we were able to use a more recent dataset and not rely on the outcomes of operations performed early in the NJR which may not reflect the current postoperative mortality trends, while still having sufficient events to be confident in our findings. The completeness of the NJR data is high. A recent NJR audit of procedure recording compliance found capture rates were 95.7% for primary procedures [22]. This reduces the likelihood of differential reporting which may

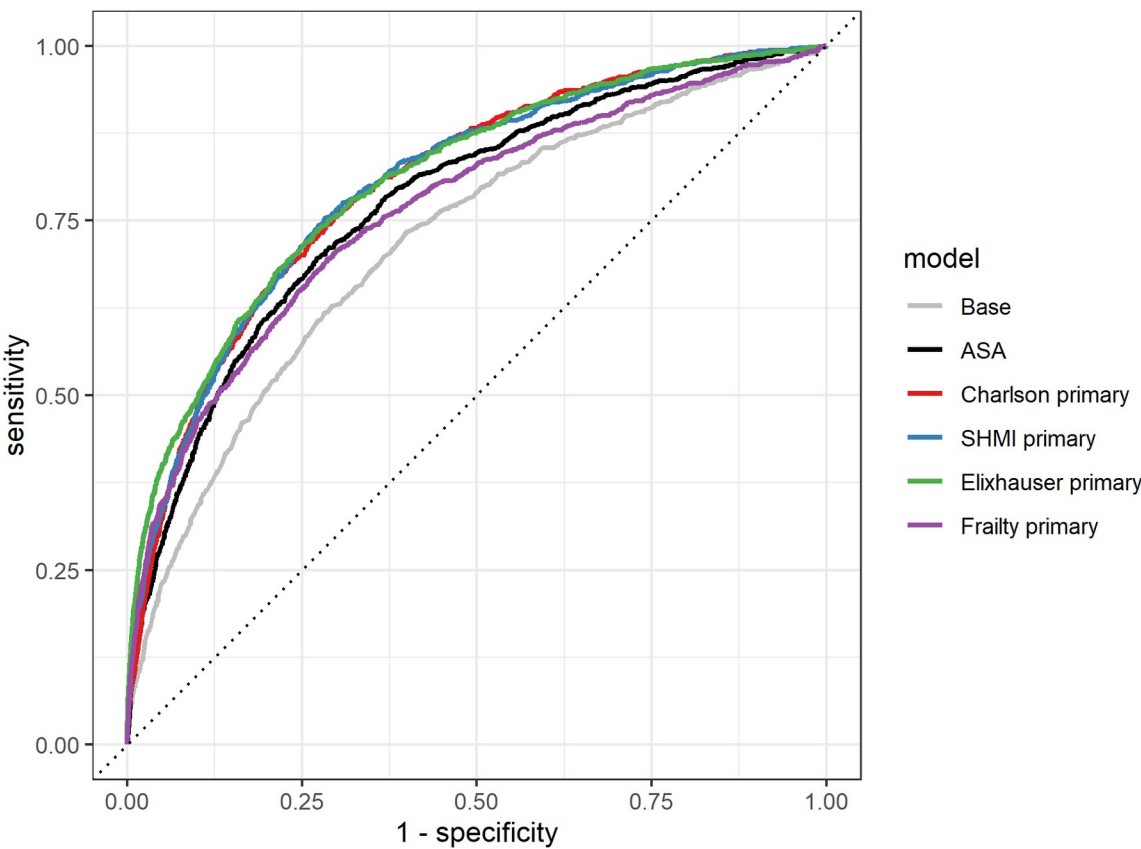

**Fig 5. ROC curves for models of 90-day mortality after THR with base model, ASA grade and the 4 comorbidity indices derived using conditions recorded at the time of the primary, including age and gender.**

have affected our models. Our ability to link with the HES data enabled us to derive four different comorbidity indices from the underlying ICD-10 codes and could potentially facilitate the derivation of more comorbidity scores in future.

The need for linkage to HES to derive comorbidity indices is also an important limitation of this study. The availability of HES data for linkage is variable, particularly for privately funded hospital episodes. Therefore, we were not able to derive comorbidity scores for many of the people who had privately funded joint replacements. These patients may have had fewer comorbidities, since private sector units tend to treat patients with fewer comorbidities than publicly funded units [23], although this may not have affected our findings. A further weakness of the HES data is that we do not know whether all pre-existing conditions are recorded for each episode, whether they are recorded accurately or whether incentives to report comorbidities have changed over time. A comparison of comorbidities recorded through HES with those from primary care records (clinical practice research database, CPRD) found that CPRD recorded more comorbidity than HES, but this did not adversely affect their models of mortality risk after gastrointestinal bleeding or diabetes [24]. This suggests that our HES records are likely to be missing some comorbidities, but these may not be important for modelling mortality risk. Some of the conditions recorded at the time of the primary operation may have been conditions which were not present on admission (i.e. complications) [25]. Our models of the risk of mortality may be missing important predictors. This study focussed on assessing whether comorbidity scores should be used instead of ASA grade in existing models, rather

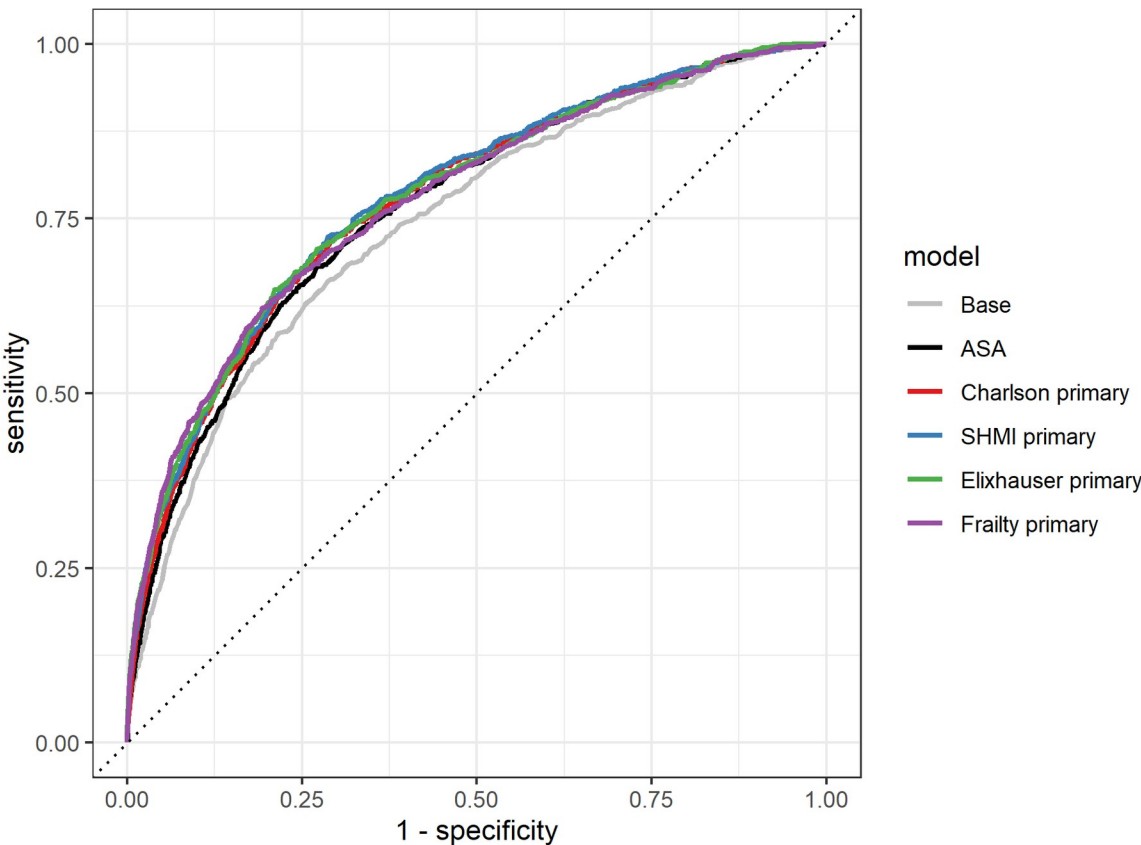

**Fig 6. ROC curves for models of 90-day mortality after KR with the base model, ASA grade and the 4 comorbidity indices derived using conditions recorded at the time of the primary, including age and gender.**

than building more comprehensive models to predict these outcomes. In future it may be valuable to consider which other variables should be included in these models. We treated some of the comorbidity scores as continuous variables and alternative parameterisations may be useful, however categorisation of continuous variables rarely increases the ability to detect differences. Although completeness of the NJR and linked mortality data are high, we do not know how many patients have missing dates of death, which may occur for example if someone emigrates after their primary operation. Given the study population and short follow-up time this is unlikely to change our main findings. Finally, we did not validate our models using an external dataset. This would be essential if we intended to develop new prediction models to be applied to new patients, but this is outside the scope of our study.

The performance of our models including CCI and Elixhauser indices (AUROC = 0.78–0.81) predicted 90-day mortality slightly worse than those by Menendez et al. (AUROC = 0.83–0.86) [5] and are comparable with those by Inacio et al. ($AUC_{THR}$ = 0.79–0.80, $AUC_{KR}$ = 0.77) [7]. The timeframe for deriving comorbidity made little difference to model performance. The modest improvements in model fit, which is consistent with Bülow et al. [10], suggest that conditions recorded at the time of the primary joint replacement operation are likely sufficient for capturing comorbidities related to post-operative mortality.

Our models predicted earlier mortality risk better than one-year mortality risk. This is unsurprising given that ASA Grade, CCI and Elixhauser index were derived to better inform risk of death or adverse events during or immediately after surgery. Bülow et al. [10] found

**Table 4. The area under the ROC curve and IPA for ASA grade and all comorbidity scores for models of 30, 45, 90, 120 and 365-day mortality after THR and KR, adjusted for age and gender.**

| Model | THRs | | | KRs | | |
|---|---|---|---|---|---|---|
| | AUROC[1] | 95% CI[2] | IPA[3] | AUROC[1] | 95% CI[2] | IPA[3] |
| **Base** | | | | | | |
| 30 days | 0.761 | 0.739–0.783 | 0.0029 | 0.756 | 0.733–0.779 | 0.0023 |
| 45 days | 0.739 | 0.719–0.760 | 0.0029 | 0.747 | 0.726–0.768 | 0.0026 |
| 120 days | 0.713 | 0.698–0.729 | 0.0042 | 0.741 | 0.726–0.756 | 0.0043 |
| 365 days | 0.708 | 0.699–0.717 | 0.0095 | 0.735 | 0.726–0.744 | 0.0103 |
| **ASA** | | | | | | |
| 30 days | 0.795 | 0.774–0.817 | 0.0038 | 0.771 | 0.749–0.793 | 0.0031 |
| 45 days | 0.781 | 0.761–0.800 | 0.0040 | 0.763 | 0.742–0.784 | 0.0037 |
| 120 days | 0.773 | 0.759–0.786 | 0.0078 | 0.763 | 0.748–0.778 | 0.0064 |
| 365 days | 0.755 | 0.746–0.763 | 0.0175 | 0.756 | 0.747–0.764 | 0.0142 |
| **CCI (original)** | | | | | | |
| 30 days | 0.817 | 0.797–0.837 | 0.0060 | 0.785 | 0.763–0.807 | 0.0049 |
| 45 days | 0.810 | 0.792–0.827 | 0.0062 | 0.776 | 0.755–0.796 | 0.0052 |
| 120 days | 0.798 | 0.786–0.811 | 0.0098 | 0.772 | 0.757–0.786 | 0.0075 |
| 365 days | 0.770 | 0.762–0.778 | 0.0179 | 0.758 | 0.750–0.767 | 0.0142 |
| **CCI (SHMI)** | | | | | | |
| 30 days | 0.820 | 0.800–0.840 | 0.0056 | 0.793 | 0.771–0.814 | 0.0081 |
| 45 days | 0.813 | 0.794–0.831 | 0.0066 | 0.783 | 0.763–0.804 | 0.0085 |
| 120 days | 0.798 | 0.785–0.811 | 0.0096 | 0.777 | 0.763–0.792 | 0.0107 |
| 365 days | 0.768 | 0.760–0.776 | 0.0189 | 0.760 | 0.751–0.768 | 0.0166 |
| **Elixhauser** | | | | | | |
| 30 days | 0.828 | 0.808–0.848 | 0.0139 | 0.791 | 0.770–0.813 | 0.0071 |
| 45 days | 0.822 | 0.804–0.840 | 0.0151 | 0.783 | 0.762–0.803 | 0.0078 |
| 120 days | 0.803 | 0.790–0.816 | 0.0212 | 0.778 | 0.763–0.793 | 0.0112 |
| 365 days | 0.770 | 0.762–0.779 | 0.0296 | 0.758 | 0.750–0.767 | 0.0181 |
| **Frailty** | | | | | | |
| 30 days | 0.808 | 0.786–0.829 | 0.0032 | 0.794 | 0.772–0.817 | 0.0003 |
| 45 days | 0.792 | 0.772–0.812 | 0.0032 | 0.786 | 0.765–0.807 | 0.0009 |
| 120 days | 0.756 | 0.741–0.772 | 0.0074 | 0.772 | 0.756–0.787 | 0.0080 |
| 365 days | 0.731 | 0.722–0.740 | 0.0167 | 0.749 | 0.741–0.758 | 0.0148 |

All comorbidity scores were derived using conditions recorded at the time of the primary operation.

1 –AUROC–area under the ROC curve.

2–95% CI– 95% confidence intervals.

3 –IPA–Index of Prediction Accuracy.

that, while comorbidity score (Elixhauser or Charlson) on its own was a poor predictor of mortality risk 5–14 years after primary THR, the performance of models which included age and gender was comparable with those for our much shorter time frame (AUROC = 0.74–0.76). This indicates that the decrease in discriminative ability we observed for models of 365-day mortality compared with 30-day mortality may plateau for risk of mortality beyond one year.

This research has confirmed, using a very large national dataset with very good coverage and completeness, that there is little advantage to using comorbidity scores rather than ASA grade to predict risk of mortality within one year of elective hip and knee replacement. Future

research may explore whether these models can be improved by using other algorithms in addition to logit models, particularly for very rare outcomes such as mortality after elective replacement. However, logit models are generally considered to be robust and perform well. Although we have used the comorbidity indices as they have been used in many other studies, the additive approach used to combine conditions in the CCI is algebraically incorrect [26] and Elixhauser et al. intended the comorbidities to be retained as independent measures rather than used to derive a summary Elixhauser index [8]. It may therefore be valuable to explore the impact of these accepted but incorrect approaches may have on mortality prediction. Finally, it may be beneficial to investigate whether comorbidity scores or specific comorbid conditions predict risk of revision after joint replacement surgery.

## Conclusions

The comorbidity scores used in this study offered little to no improvements over ASA grade in models of mortality between 30 and 365 days after elective hip or knee replacement surgery. If ASA grade is already available and linkage between datasets is needed to derive comorbidity scores, the inability to link some operations and the additional technical and administrative burdens of including comorbidity scores in models of mortality are not justified.

## Supporting information

**S1 Table. ASA grade and comorbidity scores for the study sample of people having a primary THR and KR.**
(DOCX)

**S2 Table. A comparison of the comorbidity scores of people having a primary THR who died within 90-days of their operation and those who were alive at 90-days.**
(DOCX)

**S3 Table. A comparison of the comorbidity scores of people having a primary KR who died within 90-days of their operation and those who were alive at 90-days.**
(DOCX)

**S4 Table. The area under the ROC curve and IPA scores from each of the 5 cross-validation folds for ASA grade and all comorbidity scores for models of 90-day mortality after THR, adjusted for age and gender.**
(DOCX)

**S5 Table. The area under the ROC curve and IPA scores from each of the 5 cross-validation folds for ASA grade and all comorbidity scores for models of 90-day mortality after KR, adjusted for age and gender.**
(DOCX)

**S1 Fig.** Histograms comparing the distribution of Charlson Comorbidity Index (original, blue) calculated over different lead-up times, compared with using all episodes (grey).
(DOCX)

**S2 Fig.** Histograms comparing the distribution of Charlson Comorbidity Index (SHMI, blue) calculated over different lead-up times, compared with using all episodes (grey).
(DOCX)

**S3 Fig.** Histograms comparing the distribution of Elixhauser comorbidity scores (blue) calculated over different lead-up times, compared with using all episodes (grey).
(DOCX)

**S4 Fig.** Histograms comparing the distribution of HFRS (blue) calculated over different lead-up times, compared with using all episodes (grey).
(DOCX)

**S5 Fig. A comparison of ROC curves from logit models of 90-day mortality after primary THR and ASA + Charlson comorbidity scores derived using different lead-up periods.**
(DOCX)

**S6 Fig. A comparison of ROC curves from logit models of 90-day mortality after primary THR and ASA + SHMI comorbidity scores derived using different lead-up periods.**
(DOCX)

**S7 Fig. A comparison of ROC curves from logit models of 90-day mortality after primary THR and ASA + Elixhauser comorbidity scores derived using different lead-up periods.**
(DOCX)

**S8 Fig. A comparison of ROC curves from logit models of 90-day mortality after primary THR and ASA + HFRS comorbidity scores derived using different lead-up periods.**
(DOCX)

**S9 Fig. A comparison of ROC curves from logit models of 90-day mortality after primary KR and ASA + Charlson comorbidity scores derived using different lead-up periods.**
(DOCX)

**S10 Fig. A comparison of ROC curves from logit models of 90-day mortality after primary KR and ASA + SHMI comorbidity scores derived using different lead-up periods.**
(DOCX)

**S11 Fig. A comparison of ROC curves from logit models of 90-day mortality after primary KR and ASA + Elixhauser comorbidity scores derived using different lead-up periods.**
(DOCX)

**S12 Fig. A comparison of ROC curves from logit models of 90-day mortality after primary KR and ASA + HFRS comorbidity scores derived using different lead-up periods.**
(DOCX)

**S13 Fig. A comparison of ROC curves from logit models of 30, 45, 120 and 365-day mortality after primary THR and ASA grade and all comorbidity scores derived using conditions recorded at the time of the primary.**
(DOCX)

**S14 Fig. A comparison of ROC curves from logit models of 30, 45, 120 and 365-day mortality after primary KR and ASA grade and all comorbidity scores derived using conditions recorded at the time of the primary.**
(DOCX)

## Acknowledgments

We thank the patients and staff of all the hospitals who have contributed data to the National Joint Registry. We are grateful to the Healthcare Quality Improvement Partnership, the National Joint Registry Steering Committee, and staff at the National Joint Registry for facilitating this work.

## Author Contributions

**Conceptualization:** Chris M. Penfold, Michael R. Whitehouse, Ashley W. Blom, Andrew Judge, J. Mark Wilkinson, Adrian Sayers.

**Data curation:** Adrian Sayers.

**Formal analysis:** Chris M. Penfold, Adrian Sayers.

**Funding acquisition:** Ashley W. Blom.

**Methodology:** Chris M. Penfold, Michael R. Whitehouse, Ashley W. Blom, Andrew Judge, Adrian Sayers.

**Writing – original draft:** Chris M. Penfold.

**Writing – review & editing:** Chris M. Penfold, Michael R. Whitehouse, Ashley W. Blom, Andrew Judge, J. Mark Wilkinson, Adrian Sayers.

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
