## [Decision Letter · Decision Letter 0]

12 May 2021

PONE-D-21-03064

A comparison of comorbidity measures for predicting mortality after elective hip and knee arthroplasty: A cohort study of data from the National Joint Registry in England and Wales

PLOS ONE

Dear Dr. Penfold,

Thank you for submitting your manuscript to PLOS ONE. After careful consideration, we feel that it has merit but does not fully meet PLOS ONE’s publication criteria as it currently stands. Therefore, we invite you to submit a revised version of the manuscript that addresses the points raised during the review process.

We look forward to receiving your revised manuscript.

Kind regards,

Osama Farouk

Academic Editor

PLOS ONE

Journal Requirements:

3) Please include captions for your Supporting Information files at the end of your manuscript, and update any in-text citations to match accordingly. Please see our Supporting Information guidelines for more information: http://journals.plos.org/plosone/s/supporting-information.

4) As part of your revisions, we require that you revise your ethics statement so that you indicate (1) whether data within the database are anonymized/de-identified; (2) the steps that you undertook to access the database; (3) whether you contacted your ethics oversight committee to obtain a waiver and/or to have a discussion about the study prior to accessing the database. (4) Please re-write the sentence that reads: "According to the specifications of the NHS Health Research Authority, separate informed consent and ethical approval were not required for the present study," to also  include a link to the NHS Health Research Authority's website to cite such specifications.

5) Your ethics statement should only appear in the Methods section of your manuscript. If your ethics statement is written in any section besides the Methods, please move it to the Methods section and delete it from any other section. Please ensure that your ethics statement is included in your manuscript, as the ethics statement entered into the online submission form will not be published alongside your manuscript.

6)  We note that you have indicated that data from this study are available upon request. PLOS only allows data to be available upon request if there are legal or ethical restrictions on sharing data publicly. For information on unacceptable data access restrictions, please see http://journals.plos.org/plosone/s/data-availability#loc-unacceptable-data-access-restrictions.

7) We note that the grant information you provided in the ‘Funding Information’ and ‘Financial Disclosure’ sections do not match.

8) Thank you for stating the following in the Competing Interests section:

[I have read the journal's policy and the authors of this manuscript have the following

competing interests: MW (Stryker, Heraeus, DePuy), AB (Stryker) and JMW (Amgen)

have received research and other financial support from companies or suppliers

outside the submitted work. AJ declares advisory board positions with receipt of fees

(Anthera Pharmaceuticals, INC.) and paid consultancy work (Freshfields Bruckhaus

Deringer) for companies outside the submitted work. MRW (Hip International) and

JMW (Bone and Joint Research, Journal of Orthopaedic Research) declare journal

editorial positions. JMW is a board member for the British Orthopaedic Research

Society. All other authors declare no competing interests.].

Reviewers' comments:

Reviewer's Responses to Questions

**Comments to the Author**

1. Is the manuscript technically sound, and do the data support the conclusions?

Reviewer #1: Yes

Reviewer #2: Yes

2. Has the statistical analysis been performed appropriately and rigorously? 

Reviewer #1: Yes

Reviewer #2: Yes

3. Have the authors made all data underlying the findings in their manuscript fully available?

Reviewer #1: No

Reviewer #2: Yes

4. Is the manuscript presented in an intelligible fashion and written in standard English?

Reviewer #1: Yes

Reviewer #2: Yes

5. Review Comments to the Author

Reviewer #1: The article concludes that standard comorbidity measures are not needed to predict mortality after elective arthroplasty surgery of the hip and knee. The result is coherent with previously published result but provides additional certainty due to the large sample size with high quality data. The result is important since there is a common misconception that comorbidity (as captured from administrative data) should always be used in prediction models of mortality after such surgery. It is showed in the paper that a simpler model with age, sex and ASA class is good enough, and would therefore be preferred due to simplicity.

I have some comments:

-----------------

1. It might be interesting to compare the models with a base model only. Hence, a model with age and sex only. I suspect that such model would not perform much worse than model 1?

2. I am a little skeptical to the use of the Brier score. It is a common measure of discriminatory ability, but it is actually quite insensitive for rare events data, as seen from the tables where this metric is almost constant. I would consider omitting this metric. A near-by-alternative might be the Index of Prediction Accuracy (IPA) instead. It is still based on the Brier score but with a better scale.

3. I agree that external validation seems to be outside the scope of the article. The data set is quite big, however, so I am just wondering whether it wouldn’t be possible to save a small portion from the original data for later assessment?

4. You should probably discuss if there was any (and if so, how much) censoring, especially for the one-year time frame. If relevant, you might discuss why you did not consider a time-dependent ROC-curve.

5. I would suggest a sensitivity analysis without stratification for the bootstrap replicates used for the confidence intervals. I think you should have a reasonably good chance to succeed without it (which would be a strength).

Smaller issues

----------------

1. The use of logistic regression might be a little problematic. Nevertheless, I would use it myself (I am not saying I know a better method!), but it might be discussed as a limitation that this method is actually not recommended for very rare outcomes (which is what we have here).

2. I am lacking some details of how you treated patients with surgeries performed on multiple joints. They can only die once I think.

3. Is it reasonable that patients with ASA class V get elective surgery?

4. You refer to the comorbidity indices as being “continuous”. I guess this might be a quite reasonable approximation for the Elixhauser score (which is on a discrete scale from 0 to 30). I am less convinced for CCI, however, since it is a weighted sum with only a limited set of possible values.

5. There has been some (relatively recent and perhaps valid) criticism concerning the weights used in CCI. Might be worth considering? See https://www.jclinepi.com/article/S0895-4356(16)30676-X/pdf

6. It might be noted that Elixhauser herself did not recommend using her tool as a combined measure. Therefore, it might be relevant to point out that what is not working, is to use her tool as is commonly done in practice, although usage according to her original suggestion might still work (which is usually not possible, however, due to limited data and the need of too many variables).

7. I would prefer to have some data on ASA class (and perhaps diagnosis) already in table 1 and 2 (it is now in the supplement).

8. It might be beneficial to instead include some additional data concerning individual comorbidities (according to CCI/Elixhauser) in the supplement. It might be slightly outside the scope of the article, but I do think it would be interesting to be able to compare conditions from the UK to similar data from other countries (as from other papers).

Tiny issues

----------

9. Abstract: I propose to describe the data as “linked” rather than “combined”.

10. Abstract: “Mortality _after_ 90 days”. Shouldn’t this be “Mortality _within_ 90 days”?

11. I am not a native English speaker (and I know you are) but for me “sex” would sound more accurate than “gender”, since you use administrative register data. Or were the patients surveyed for gender identity? (You do actually have “sex” in fig 5.)

12. I thought ASA was a “class” (not a “grade”)? (I am not sure of the difference myself but this is what I have been told).

13. Background: Isn’t Guernsey part of the register (NJR) name (it is mentioned later).

14. Background: “the main focus within joint replacement surgery has been on the Charlson Comorbidity (CCI) and Elixhauser indices”. I think the RxRisk score has been used quite a bit as well (at least in Australia). Could perhaps be mentioned?

15. You state that the “original” Elixhauser includes 30 conditions. This is true but since it originally included 31, it might be a little confusing what is actually meant by “original” (one condition was dropped in an update made in 2004).

16. Methods: Was “HES” used for Wales as well?

17. Methods: I think that what you refer to as “predictors” might be considered “potential predictors” or similar?

18. You might spell out the meaning of “ICD-10” the first time you use this acronym.

19. Predictors section: What is meant by “All available episodes up to the primary”. I suppose that the start of HES indicates a lower limit. When was that?

20. There is a mention of a Brier score of “0.0.0034” (two decimal marks).

21. You mention as a strength that you do not need to rely on data that is older than 10 years (and I agree!). But you do in fact include data from early 2011 (which strictly speaking was more than ten years ago).

22. You mention that HES data from private hospitals were not included and that those patients might be healthier than other patients. I agree, but I also suppose that in some other countries, private hospitals might also have other incitements to report additional comorbidities due to reimbursement? But I suppose you do not have that issue in the UK?

23. Figure 1 and 2: I would prefer dates as “2011-01-01” (month before day since I think this is more common in other countries.

24. Table 1 and 2 have some superscripts which are not explained.

25. For figures S5 etc I think you should clarify that age and sex were included in the models (otherwise the ROC curves should be less smooth).

26. I think you have too many digits in your confidence intervals for AUC. If you have “only” 2000 bootstrap replicates you should only include one significant digit. If you need two digits, you are recommended to have 10000 replicates (according to the documentation for the pROC R package).

27. I would recommend including pagination and line numbers for the sake of the reviewers :-)

Reviewer #2: - I think it would be better and more consistent to use arthroplasty instead of replacement consistently throughout the whole article and consequently using the abbreviations THA and TKA rather than using both terms at different sections of the article.

- I would suggest pointing out the most commonly encountered specific comorbidities which might have been correlating / predicting mortality after THA or TKA ...It would have ideal to include them in the analysis in addition to the use of the whole scores parameters.

6. PLOS authors have the option to publish the peer review history of their article (what does this mean?). If published, this will include your full peer review and any attached files.

Reviewer #1: **Yes: **Erik Bülow

Reviewer #2: **Yes: **Mahmoud Abdel Karim

---

## [Author Response · Author response to Decision Letter 0]

29 Jun 2021

Response to reviewers’ comments on manuscript # PONE-D-21-03064

18th June 2021

Dear Chief Editor,

Please find enclosed our revised manuscript, entitled A comparison of comorbidity measures for predicting mortality after elective hip and knee replacement: A cohort study of data from the National Joint Registry in England and Wales. 

We thank the reviewers for their careful review of our article. They have highlighted important areas in which our article can be improved. We have responded to each comment below.

We look forward to hearing from you at your earliest convenience,

Yours sincerely,

Dr Chris Penfold

MSci, PhD

 

Reviewer #1:

1. It might be interesting to compare the models with a base model only. Hence, a model with age and sex only. I suspect that such model would not perform much worse than model 1?

We have updated our analyses to include the base model (page 5, line 145), adjusted for age and gender only. We have also included the base model in our comparisons of model performance (page 7, lines 199-200).

2. I am a little skeptical to the use of the Brier score. It is a common measure of discriminatory ability, but it is actually quite insensitive for rare events data, as seen from the tables where this metric is almost constant. I would consider omitting this metric. A near-by-alternative might be the Index of Prediction Accuracy (IPA) instead. It is still based on the Brier score but with a better scale.

Thank you for this suggestion. We were not familiar with the Index of Prediction Accuracy. We have replaced the Brier score with the IPA and amended the relevant text in the Methods (page 6, lines 153-156) and Results (page 7, lines 200-204 and 209-214).

3. I agree that external validation seems to be outside the scope of the article. The data set is quite big, however, so I am just wondering whether it wouldn’t be possible to save a small portion from the original data for later assessment?

We have opted to use 5-fold cross-validation and have updated our Methods (page 5, lines 156-157) and Results to reflect this change. In the Results section we have included details of the change in AUC and IPA estimates within each of the 5 folds (page 8, lines 225-229 and Tables S4 and S5).

4. You should probably discuss if there was any (and if so, how much) censoring, especially for the one-year time frame. If relevant, you might discuss why you did not consider a time-dependent ROC-curve.

We assume the reviewer is specifically referring to censoring due to missing dates of deaths. Since our mortality outcome is recorded through centrally collected civil registration records, missing dates of deaths are very unlikely to occur for people still resident in the UK when they died. However, it is plausible that the recording of dates of deaths may be delayed or missed entirely for people who died after emigrating from the UK. Censoring due to loss-to-follow-up of this type has not been quantified for the NJR, but we anticipate it affecting only a small number of people and consequently having only a negligible effect on our study. We have acknowledged this as a weakness of our study (page 9, lines 272-276).

5. I would suggest a sensitivity analysis without stratification for the bootstrap replicates used for the confidence intervals. I think you should have a reasonably good chance to succeed without it (which would be a strength).

We thank the reviewer for this suggestion. Rather than include this as a sensitivity analysis we have changed our analysis approach and updated our Methods to specify that confidence intervals be calculated using the exact method to evaluate the uncertainty of AUC developed by DeLong using the algorithm proposed by Sun and Xu in the ‘pROC’ package (page 6, lines 163-165).

Smaller issues

----------------

1. The use of logistic regression might be a little problematic. Nevertheless, I would use it myself (I am not saying I know a better method!), but it might be discussed as a limitation that this method is actually not recommended for very rare outcomes (which is what we have here).

We have erred on the side of ‘tried and tested’ in our use of logistic regression, but we will keep abreast of developments in methods for very rare outcomes for future work. 

We have amended the Discussion as follows (page 10, line 298):

“Future research may explore whether these models can be improved by using other algorithms in addition to logit models, particularly for very rare outcomes such as mortality after elective arthroplasty.”

2. I am lacking some details of how you treated patients with surgeries performed on multiple joints. They can only die once I think.

A small number of patients had a second primary operation (contralateral primary) in the follow-up timeframe. We have included a sensitivity analysis in which the first primary operation for patients who had contralateral primaries was censored at the time of the 2nd primary. We have updated the Methods (page 5, lines 159-161) and Results (page 8, lines 230-232) with the relevant details. Outcomes were unaffected by this sensitivity analysis.

3. Is it reasonable that patients with ASA class V get elective surgery?

We agree that it is extremely unlikely that a patient with ASA class V would have elective surgery. A potential explanation is a simple data entry error (ASA class IV was intended). For this reason, and due to the extremely low number of people classed as ASA V, we combined ASA classes IV and V in our analyses. We have clarified this in our Methods (page 4, line 116).

4. You refer to the comorbidity indices as being “continuous”. I guess this might be a quite reasonable approximation for the Elixhauser score (which is on a discrete scale from 0 to 30). I am less convinced for CCI, however, since it is a weighted sum with only a limited set of possible values.

We have now included CCI as a categorical variable in our models and updated the Methods (page 4, lines 121) and Results to reflect this.

5. There has been some (relatively recent and perhaps valid) criticism concerning the weights used in CCI. Might be worth considering? See https://www.jclinepi.com/article/S0895-4356(16)30676-X/pdf

We have included this as a discussion point. Given that the focus of our study is on whether comorbidity scores as they are currently used offer any advantage over ASA grade we feel it is beyond the scope of this article to focus on the correct mathematical handling of the weights used in CCI. However, we anticipate this will be an important area of work in the future. We have amended our Discussion to include the following (page 10, lines 299-304):

“Although we have used the comorbidity indices as they have been used in many other studies, the additive approach used to combine conditions in the CCI is algebraically incorrect (25) and Elixhauser et al. intended the comorbidities to be retained as independent measures rather than used to derive a summary Elixhauser index (8). It may therefore be valuable to explore the impact of these accepted but incorrect approaches may have on mortality prediction.”

6. It might be noted that Elixhauser herself did not recommend using her tool as a combined measure. Therefore, it might be relevant to point out that what is not working, is to use her tool as is commonly done in practice, although usage according to her original suggestion might still work (which is usually not possible, however, due to limited data and the need of too many variables).

 Please see the amendment to our Discussion summarised above (point 5).

7. I would prefer to have some data on ASA class (and perhaps diagnosis) already in table 1 and 2 (it is now in the supplement).

 We have included ASA grade in Tables 1 and 2.

8. It might be beneficial to instead include some additional data concerning individual comorbidities (according to CCI/Elixhauser) in the supplement. It might be slightly outside the scope of the article, but I do think it would be interesting to be able to compare conditions from the UK to similar data from other countries (as from other papers).

In Tables 1 and 2 we have included the prevalence of all the conditions which are used in the CCI and Elixhauser Index. We have described these in our Results (pages 6 & 7, lines 174-185), including highlighting similarities and differences in patients who die within 90 days of THA or KA.

Tiny issues

----------

9. Abstract: I propose to describe the data as “linked” rather than “combined”.

 We have replaced ‘combined’ with ‘linked’ in the Abstract.

10. Abstract: “Mortality _after_ 90 days”. Shouldn’t this be “Mortality _within_ 90 days”?

 Thanks! We have corrected this.

11. I am not a native English speaker (and I know you are) but for me “sex” would sound more accurate than “gender”, since you use administrative register data. Or were the patients surveyed for gender identity? (You do actually have “sex” in fig 5.)

Thank you for identifying this inconsistency. The NJR data collection form refers to gender, we have therefore used this in our article. We have checked and corrected the text, plots and tables where necessary.

12. I thought ASA was a “class” (not a “grade”)? (I am not sure of the difference myself but this is what I have been told).

In all honesty we aren’t sure whether ASA should be referred to as a grade or class. The unabbreviated form of ASA grade/class is “ASA physical status classification system” which suggests it should be ASA class, but the National Institute for Health and Clinical Excellence (NICE), and many other professional bodies including the NJR, refers to ‘ASA grade’. Given the lack of consensus on this, we hope the reviewer won’t mind us keeping this as ‘ASA grade’ since this is in agreement with the NJR data collection form. We are happy to include this change if necessary.

13. Background: Isn’t Guernsey part of the register (NJR) name (it is mentioned later).

 We have updated the Background to include the States of Guernsey.

14. Background: “the main focus within joint replacement surgery has been on the Charlson Comorbidity (CCI) and Elixhauser indices”. I think the RxRisk score has been used quite a bit as well (at least in Australia). Could perhaps be mentioned?

Thanks for highlighting this. We agree that RxRisk score has been used in joint replacement surgery, but it is a prescription rather than diagnosis-based score. We have revised the wording in our Background to specify that the CCI and Elixhauser index are widely used diagnosis-based summary scores.

15. You state that the “original” Elixhauser includes 30 conditions. This is true but since it originally included 31, it might be a little confusing what is actually meant by “original” (one condition was dropped in an update made in 2004).

We thank the reviewer for this comment. We have checked the 1998 paper by Elixhauser et al (cited in our article), which to our knowledge is the original source for the Elixhauser index, and it refers to 30 conditions. We may have overlooked an earlier article and we would be very happy to update this description if the reviewer can direct us to the relevant literature. 

16. Methods: Was “HES” used for Wales as well?

HES data are only available for England. We have clarified this within our definition of the study sample (page 4, lines 107-108).

17. Methods: I think that what you refer to as “predictors” might be considered “potential predictors” or similar?

 We have replaced “predictors” with “potential predictors”.

18. You might spell out the meaning of “ICD-10” the first time you use this acronym.

 We have included ICD-10 in our list of abbreviations. Sorry for this oversight.

19. Predictors section: What is meant by “All available episodes up to the primary”. I suppose that the start of HES indicates a lower limit. When was that?

 We have updated our Methods to clarify that HES records are available from 1989 onwards (page 4, line 98).

20. There is a mention of a Brier score of “0.0.0034” (two decimal marks).

 We have replaced the Brier score with the IPA.

21. You mention as a strength that you do not need to rely on data that is older than 10 years (and I agree!). But you do in fact include data from early 2011 (which strictly speaking was more than ten years ago).

 We have amended the wording as follows (page 8, line 246):

“The size of the NJR meant that we were able to use a more recent dataset and not rely on the outcomes of operations performed early in the NJR which may not reflect the current postoperative mortality trends, while still having sufficient events to be confident in our findings.”

22. You mention that HES data from private hospitals were not included and that those patients might be healthier than other patients. I agree, but I also suppose that in some other countries, private hospitals might also have other incitements to report additional comorbidities due to reimbursement? But I suppose you do not have that issue in the UK?

We are not aware of evidence to support incitement to report additional comorbidities in the UK and have been looking into this in related work. We have amended our discussion of weaknesses to include this possibility alongside other discussions around the recording of comorbidities in HES (page 9, line 260).

23. Figure 1 and 2: I would prefer dates as “2011-01-01” (month before day since I think this is more common in other countries.

 We have amended the date format in Figures 1 and 2

24. Table 1 and 2 have some superscripts which are not explained.

Thank you and sorry for this oversight. We have included footnotes to explain the superscripts in Tables 1 and 2.

25. For figures S5 etc I think you should clarify that age and sex were included in the models (otherwise the ROC curves should be less smooth).

 We have amended the figure titles to include reference to age and gender being in the models.

26. I think you have too many digits in your confidence intervals for AUC. If you have “only” 2000 bootstrap replicates you should only include one significant digit. If you need two digits, you are recommended to have 10000 replicates (according to the documentation for the pROC R package).

Given the revised calculation of the confidence intervals in response to point 5 in the ‘Main issues’ we have retained the current number of significant figures for the confidence intervals.

27. I would recommend including pagination and line numbers for the sake of the reviewers :-)

Sorry for omitting these in the original submission. We’ve added them in to the revised version, which will hopefully make life a bit easier!

Reviewer #2: 

1. I think it would be better and more consistent to use arthroplasty instead of replacement consistently throughout the whole article and consequently using the abbreviations THA and TKA rather than using both terms at different sections of the article.

Thank you for highlighting this inconsistency. The NJR uses replacement rather than arthroplasty and since we are specifically referring to joint replacements rather than other procedures captured by arthroplasty we have updated our article to use replacement throughout. 

2. I would suggest pointing out the most commonly encountered specific comorbidities which might have been correlating / predicting mortality after THA or TKA ...It would have ideal to include them in the analysis in addition to the use of the whole scores parameters.

We have included the prevalence of the underlying conditions used to calculate the Charlson and Elixhauser scores in Tables 1 and 2. We have also described the most prevalent comorbidities in patients who died within 90 days of their primary THR or KR (pages 6 & 7, lines 174-185). We agree that it may be interesting to include the individual conditions in our analyses, particularly to explore their validity in predicting mortality risk for arthroplasty patients. However, the focus of this article is on the use of existing comorbidity scores compared with ASA grade. Reviewer 1 raised the point that Elixhauser did not intend for individual comorbidities to be combined into a summary score. We have included exploring the impact of retaining individual comorbidities for predicting mortality risk as an area for future research (page 10, lines 299-304).

---

## [Decision Letter · Decision Letter 1]

21 Jul 2021

A comparison of comorbidity measures for predicting mortality after elective hip and knee replacement: A cohort study of data from the National Joint Registry in England and Wales

PONE-D-21-03064R1

Dear Dr. Penfold,

We’re pleased to inform you that your manuscript has been judged scientifically suitable for publication and will be formally accepted for publication once it meets all outstanding technical requirements.

Kind regards,

Osama Farouk

Academic Editor

PLOS ONE

Additional Editor Comments (optional):

Reviewers' comments:

Reviewer's Responses to Questions

**Comments to the Author**

1. If the authors have adequately addressed your comments raised in a previous round of review and you feel that this manuscript is now acceptable for publication, you may indicate that here to bypass the “Comments to the Author” section, enter your conflict of interest statement in the “Confidential to Editor” section, and submit your "Accept" recommendation.

Reviewer #1: All comments have been addressed

Reviewer #2: All comments have been addressed

2. Is the manuscript technically sound, and do the data support the conclusions?

Reviewer #1: Yes

Reviewer #2: Yes

3. Has the statistical analysis been performed appropriately and rigorously? 

Reviewer #1: Yes

Reviewer #2: I Don't Know

4. Have the authors made all data underlying the findings in their manuscript fully available?

Reviewer #1: Yes

Reviewer #2: No

5. Is the manuscript presented in an intelligible fashion and written in standard English?

Reviewer #1: Yes

Reviewer #2: Yes

6. Review Comments to the Author

Reviewer #1: I am thankful for the opportunity to review this nice paper once more. The authors have provided thorough and well-motivated answers to all my questions. I feel more than satisfied! I am looking forward to reading this article in print!

P.S. As a small side note (no action requested), I would just like to expand on my comment no. 15, regarding the number of Elixhauser comorbidities. The authors are right that Elixhauser et al. (1998) refers to 30 comorbidities (my mistake!). Those are listed in table 1 from 1998. The second comorbidity is “Cardiac arrhythmias” and the 6:th is “Hypertension (combined)”. Some later papers (for example Quan et al 2005; https://doi.org/10.1097/01.mlr.0000182534.19832.83) have split hypertensions to either “uncomplicated” or ”complicated”. Hence, with a total of 31, while still refereeing to the original paper.

In addition to the published papers, The Elixhauser comorbidity software is provided by the AHRQ (https://www.hcup-us.ahrq.gov/toolssoftware/comorbidity/comorbidity.jsp#archives). It was described in a tech note from 2004 that “[Cardiac arrhythmias] was removed for FY2004, Version 2.0.” (p 6; https://www.hcup-us.ahrq.gov/toolssoftware/comorbidity/Table1-FY2004-V2_1.pdf). Thus, leaving 29 comorbidities for later versions of the software.

A rhetorical question (no need for explicit response): How many Elixhauser comorbidities are there in table 1 and 2 of the now revised manuscript? 

Reviewer #2: Thank you so much for replying to the comments and suggestions raised. I think you have managed to address all of them.

Thank you for your efforts and interesting article.

7. PLOS authors have the option to publish the peer review history of their article (what does this mean?). If published, this will include your full peer review and any attached files.

Reviewer #1: **Yes: **Erik Bülow

Reviewer #2: **Yes: **Mahmoud Abdel Karim

---

## [Editor Report · Acceptance letter]

3 Aug 2021

PONE-D-21-03064R1 

A comparison of comorbidity measures for predicting mortality after elective hip and knee replacement: A cohort study of data from the National Joint Registry in England and Wales 

Dear Dr. Penfold:

I'm pleased to inform you that your manuscript has been deemed suitable for publication in PLOS ONE. Congratulations! Your manuscript is now with our production department. 

Kind regards, 

on behalf of

Dr. Osama Farouk 

Academic Editor

PLOS ONE